# Cervical Cancer Screening among Female Refugees in Jordan: A Cross-Sectional Study

**DOI:** 10.3390/healthcare10071343

**Published:** 2022-07-20

**Authors:** Nadia Muhaidat, Mohammad A. Alshrouf, Roaa N. Alshajrawi, Zeina R. Miqdadi, Raghad Amro, Abedalaziz O. Rabab’ah, Serena A. Qatawneh, Alaa M. Albandi, Kamil Fram

**Affiliations:** 1Department of Obstetrics & Gynaecology, School of Medicine, The University of Jordan, Amman 11942, Jordan; ruaaalshjrawi01@gmail.com (R.N.A.); zmiqdadi@gmail.com (Z.R.M.); albandialaa@gmail.com (A.M.A.); kamilfram@gmail.com (K.F.); 2Department of Surgery, Faculty of Medicine, Mutah University, Kerak 61710, Jordan; raghadamr7@gmail.com (R.A.); serenaqatawneh2002@gmail.com (S.A.Q.); 3Faculty of Medicine, Jordan University of Science and Technology, Irbid 22110, Jordan; abedalazizrababiah@gmail.com

**Keywords:** awareness, cervical cancer, human papillomavirus, cervical smear, cervical cancer screening, risk factors, refugee

## Abstract

Background: Cervical cancer (CC) is mainly linked to infection with a high-risk oncogenic human papillomavirus (HPV), with 85% of deaths occurring in developing countries. Refugees are less likely to be aware of screening methods and to have routine gynecological examinations. Methods: This is a cross-sectional study involving a total of 359 women aged 19–64 living in the Jerash camp in Jordan. Data were collected using a carefully developed and validated questionnaire. Results: A total of 359 participants were included in the study, with a mean age of 38.99 ± 10.53. Participants demonstrated fair knowledge of CC risk factors (4.77 ± 2.85 out of 11). Among the participants, 73.5% had heard of the cervical smear test; however, only 12.8% had actually undergone the test, with a mean total number of smear tests performed of 1.48 ± 0.79 and the mean age at the time of the first test was 32.5 ± 7.89. Conclusions: Refugee women have a fair level of knowledge of CC risk factors but are unmotivated to have a Pap smear test to screen for CC. Efforts should be made to raise awareness about the issue and promote testing for underrepresented women in refugee camps.

## 1. Introduction

Cervical cancer is the fourth most prevalent malignancy in women and has a high mortality rate worldwide [1]. Most cases are linked to high-risk human papillomavirus (HPV) infection, which causes cellular changes that lead to cancer [2]. These changes can be screened for by using a cervical smear test, which can facilitate the prevention or early detection of the development of cancer [3]. Therefore, the World Health Organization (WHO) cervical cancer guidelines recommend the conventional Papanicolaou test as a routine screening test for cervical cancer in females [4]. Co-testing with Pap smear and HPV testing every five years is recommended for women aged 30 to 65 due to its better sensitivity, whereas Pap smear testing is indicated for women aged 21 to 30 due to HPV testing’s lower specificity in this population [5]. In addition, the American Cancer Society recommends that all women begin cervical cancer screening at the age of 21 [6]. 

The mortality rates have declined significantly with the introduction of cervical cancer screening tools [7]. Unfortunately, developing countries continue to have lower cervical cancer screening rates than developed countries [8], and as a result, developing regions account for nearly 85% of cervical cancer deaths [9]. Furthermore, screening rates varied by country of origin; a study of immigrants in Los Angeles found that Laotians, Cambodians, Vietnamese, Asian Indian, and Chinese immigrants had lower screening rates (52–56%) than Korean, Filipino, and Japanese women (65–75%) [10]. Multiple studies showed that Arab refugees and migrants had fewer gynecological examinations than native populations, resulting in higher mortality due to delayed diagnosis [11,12].

The National Comprehensive Cancer Network (NCCN) 2019 guidelines describe early-stage disease (stages I–II) treatment options, including surgery and concurrent chemoradiation, which can result in cures in 80% of patients [13]. According to the International Federation of Gynecology and Obstetrics (FIGO), treatments are dependent on the stage, and surgery is usually only used for early-stage disease, fertility preservation, and smaller lesions, such as stage IA, IB1, and selected IIA1 cases [14]. Cervical cancer stages IB–IIA can be treated with either radical surgery or radiotherapy, and the survival outcomes are the same for both [15]. In more advanced cases, concurrent chemoradiation using cisplatin alone or a cisplatin and fluorouracil combination is the treatment of choice for stages IB2, II, III, and IVA disease [13].

In Jordan, there is no national screening program for cervical cancer; therefore, raising women’s awareness is crucial to reducing the morbidity and mortality of CC. There have been few studies on women’s cervical smear test knowledge in Jordan; a cross-sectional study of 500 age-eligible women revealed that only 31.2% had been screened for cervical cancer [16]. Another study of 8333 women discovered that only 25.5% had ever had one in their lives [17]. 

Despite several studies that have been conducted among various groups of Jordanian women to determine cervical cancer-protective behavioral patterns, knowledge, and beliefs [16,17,18,19], no study has been conducted on refugees in Jordan. This study aimed to look into the knowledge of cervical cancer signs, symptoms, risk factors, and cervical smear-seeking practices among female Palestinian refugees in the Jerash camp. In addition, it aimed to identify the barriers and facilitators for cervical cancer screening. The opportunity was also taken to raise awareness about cervical cancer risk factors and screening among the study population.

## 2. Materials and Methods

### 2.1. Study Design and Population

A cross-sectional study was conducted among Palestinian refugee women residing in the Jerash camp, Jordan. The Jerash refugee camp, which houses 37,000 refugees, is one of Jordan’s ten United Nations Relief and Works Agency (UNRWA)-recognized Palestinian refugee camps [20]. The inclusion criteria were refugees living in Jerash camp who were over the age of 19 and under the age of 65. Women who did not complete the questionnaire had missing answers or had a hysterectomy and were excluded from the study. 

This study was conducted in accordance with the Declaration of Helsinki and was granted ethical approval by the Jordan University Hospital ethics committee. Informed written consent was obtained from all the patients.

### 2.2. Sample Size and Sampling Procedures

Data were collected using a non-probability convenience sampling method from October 2021 to February 2022. The questionnaire was self-administered for literate women, with clear instructions provided prior to administration, and they had the option of filling it out on printed paper or filling out an electronic survey using an electronic device. In the case of illiterate women, an interviewer-administered approach was used. After filling out the questionnaire, the participants were given leaflets with information about cervical cancer screening. They were also given a short lecture in simple language to help them learn more about the subject.

The sample size for the study was calculated using a formula to estimate a single population proportion. A previous study conducted on a Jordanian population yielded a knowledge score about cervical cancer screening of 61% [19]. With a 95% confidence interval and a margin of error of 5%, the required sample size was 355. Accordingly, 500 participants were approached to complete the survey. Of those, 411 agreed to participate in this study, with an 82.2% participation rate. Following that, 52 responses were excluded because they were under the age of 19 or over the age of 65 or had missing responses. Finally, 359 women were included in the analysis.

### 2.3. Data Collection Procedures

After an extensive literature review, a direct questionnaire was developed for this study [21,22,23,24]. Necessary modifications were made to the adopted questions to make the final questionnaire more culturally acceptable, given the conservative nature of the population. The adopted items in the questionnaire were translated by a specialist into Arabic as recommended by the World Health Organization. Two bilingual healthcare professionals with relevant clinical and research expertise in gynecology, public health, and survey design translated it from English to Arabic. Two other bilingual healthcare professionals back-translated it into English.

The questionnaire consists of seven sections: (1) sociodemographic factors, consisting of nine questions about age, marital status, educational status, number of children, monthly family income, occupational status, type of health insurance, age at marriage, and menstrual status; (2) knowledge about signs and symptoms of cervical cancer. It was divided into 13 phrases with three possible answers each (yes/no/I don’t know); (3) knowledge of risk factors. This section presented 11 risk factors with a 5-point Likert scale; (4) attitudes regarding Pap smear screening, which consisted of 10 factors with three choices (no, does not affect me/slightly prevents me from having the test/strongly prevents me from having the test); (5) practices regarding cervical cancer screening; (6) Pap smear test knowledge, which includes four questions about general information regarding the Pap smear test with three choices (true/false/I don’t know) and an item inquiring the source from which they heard about the test; (7) knowledge of the HPV consists of six questions with choices (true/false/I don’t know). The last two parts used a skip-logic method because they were only answered if the participant said they knew about the smear test and HPV.

For questions involving cervical cancer risk factors, responses of “strongly agree” and “agree” were deemed right, but those of “strongly disagree”, “disagree”, or “not sure” were considered wrong. The participant received one point for each correctly indicated risk factor, and a score similar to the one used in a previous study [25] was then calculated and classified into three categories: poor knowledge (0–3), fair knowledge (4–7), and good knowledge (8–11).

The reliability and validity of the questionnaire were analyzed by conducting a pilot study on 22 women who were not included in the final analysis of the study. The Cronbach’s alpha value for the knowledge about signs and symptoms of cervical cancer scale, the 5-point Likert scale regarding knowledge about risk factors, and attitudes regarding the Pap smear screening scale were 0.869, 0.734, and 0.661, respectively. Slight modifications were made to the questionnaire after the pilot study; one item was rephrased, and two items were deleted to make it easier to comprehend.

### 2.4. Statistical Analysis

SPSS version 28.0 (Chicago, IL, USA) was used in our analysis. The data were described using variability analysis in the form of means (standard deviation). The socio-demographic factors were calculated and provided as frequencies (percentages) using standard descriptive statistical parameters. The reliability of the questionnaires was computed via Cronbach’s alpha. The relationship between categorical study factors was examined using the Chi-square test and Fisher’s exact tests. Continuous variables were compared using ANOVA followed by a post hoc (LSD) test to compare means among all studied or independent sample *t*-tests as appropriate, and a Levene’s test was used to test for homogeneity of variances. Statistical significance was defined as a *p*-value of less than 0.05.

## 3. Results

### 3.1. Characteristics of the Sample

Overall, 359 women participated in this study and were included in the analysis. The participants’ ages ranged from 19 to 64, with a mean of 38.99 ± 10.53. The 30 to 40 age group was the most represented (35.2%). Most of the participants were married (78%), with an average age of marriage of 21.34 ± 5.16 years old, and most of the participants had four to six children (37.6%), with a mean number of children of 3.97 ± 2.8 (range, 0 to 11).

The participants’ educational levels were as follows: 69.6% had secondary education or higher, and most (85%) were unemployed. The majority (78%) of the participants did not have health insurance, and more than half (53%) had an income of less than 200 JD per month or had no income at all. Table 1 demonstrates the socio-demographic and Pap smear screening characteristics of participants.

### 3.2. Participants’ Knowledge of CC and HPV

The most commonly perceived symptoms of cervical cancer were a constant pain in the lower back (49.3%), bleeding after menopause (47.6%), and having more menstruations than usual (46.2%) (Table 2).

Only 13.6% of the participants had heard about HPV, and 15.3% of the participants had heard about Chlamydia bacteria. Among those who have heard about HPV, 49% believed it is an uncommon virus with a low infection rate. More than half (57.1%) of them thought that HPV is a direct cause of CC. In addition, 40.8% of them said that HPV signs and symptoms always appear when infected. Table 3 demonstrates the participants’ knowledge of HPV.

### 3.3. Attitudes and Perceptions of Participants toward Cervical Cancer Risk Factors

Table 4 shows the participants’ knowledge of cervical cancer risk factors; the average mean score was 4.77 ± 2.85 out of 11, indicating fair knowledge overall. In terms of cervical cancer risk factors, 34% had poor knowledge, 47.6% had fair knowledge, and just 18.4% had good knowledge.

### 3.4. The Participants’ Knowledge and Perceived Personal-Related Barriers toward Pap Smear

Overall, 73.5% (n = 264) had heard of the cervical smear test. More than half (59.5%) of the participants believed that all women should receive a Pap smear test every five years, and two-thirds (65.2%) agreed that the Pap smear test should be for all women, not only those at risk of cervical cancer (Table 5).

Despite this, only 12.8% of the participants had ever had a Pap smear in their lifetime, with a mean number of total smears performed of 1.48 ± 0.79. The mean age at the time of the first smear test was 32.5 ± 7.89, with a range of 21 to 50. Overall, on average, the participants underwent a Pap smear every 6.14 ± 4.83 years (range, 0 to 15). Among the 46 participants (12.8%) who had had a smear before, did so at a private hospital (43.5%), a government hospital (26.1%), the Jordanian Association for Family Planning (16.7%), a UNRWA clinic (6.5%), or somewhere else (8.7%).

Among the participants, 39.3% said they were willing to have a smear test, 30.1% were unsure, and 30.6% were unwilling. A significant difference was found between the ages of participants willing to have a smear test versus those who were not (*p* < 0.001), with a mean difference of 1.28 years higher in the willing participants (95% CI 0.58 to 1.98). In addition, women after menopause were more likely to be unwilling to have a smear (43.1%) than women before menopause (27.5%) (*p* = 0.032). Moreover, there was a significant association between marital status and willingness to have a smear test (*p* = 0.04), where 40% of the married participants said they were willing to have a smear test, compared to 27.5% among single participants.

Among the perceived barriers by the participants was the cost of the examination (67.4%), the test being performed on many occasions by a male doctor (66.9%), lack of health services nearby (64.6%), believing that their health is good and they do not need it (58.2%), and fear of knowing the result (56.3%), were the most common (Table 6). There was a significant association between the participants’ employment status and their perception of the cost as a barrier (*p* = 0.001), with 71.1% of the unemployed participants saying that the cost prevents them from undergoing the test, compared to 46.3% of the employed participants.

### 3.5. Source of Knowledge on Cervical Cancer Screening

Almost three-quarters (73.5%, n = 264) of the women had heard of the cervical smear test. Among those who heard about the cervical smear test, the participants were asked about the primary source of information and could select more than one response. It was found that awareness campaigns, such as radio, ads, and social networking sites (66.2%), were the main way people obtained the information, followed by friends or family (20.2%) and doctors (20.2%).

## 4. Discussion

Our study demonstrated a fair level of awareness of CC risk factors. Although the majority of participants had secondary education or higher, only 18.4% had good knowledge of CC risk factors, which could be attributed to stigma and a lack of involvement in such matters in educational systems. In contrast, a study of Syrian refugees in Greece indicated that having a higher level of education and money was linked to having a better level of knowledge [26]. Numerous studies have shown that education level has a significant impact on screening uptake and women’s knowledge of CC risk factors. A study about the uptake of cervical and breast cancer screening among women in Spain found that those with higher levels of education were more likely to have undergone cervical cancer screening than those with lower levels of education or only a primary education [27]. In addition, prior cross-sectional research among women in the Gaza Strip revealed that participants with only a secondary or diploma degree had a reduced likelihood of being well-versed in CC risk factors [25,28]. 

We found that the most well-known cancer risk factors among participants were low immunity and smoking, while early marriage and having more children were the least reported risk factors. These findings are similar to those of a study conducted among Palestinian women in the Gaza Strip, which is likely since most of the women in our study are Gaza refugees living in Jordan, sharing many cultural similarities with the Palestinian community [25]. In that study, it was found that the overall awareness of CC risk factors with good knowledge was 23.7%, which is similar to our study with 18.4% [25]. 

The participants’ knowledge of HPV was found to be low, with only around 13% having heard of it, compared to 59% of participants in research among young adult Italian women. This could be explained by Italy’s national public efforts to fund screening programs every three years, as well as the research projects focusing on HPV, its vaccine, and the inclusion of the HPV vaccine in the national immunization program [29]. In Jordan, the HPV vaccine is only available if one is willing to pay, and according to a 2021 study, the willingness to pay for the vaccine was only 16% [30]. Moreover, more than half (57.1%) of those who had heard of HPV knew that it is a direct cause of cervical cancer, which is more than the knowledge level reported in a study among Chinese women, which was found to be 53.4% [31].

In terms of cervical smear test knowledge, 73.5% of our study population knew about it, which is higher than a study conducted in Nepal, where only 18% knew about it [32]. However, only 12.8% of our participants had ever been screened for cervical cancer, which is comparable to India’s screening test rate of 9.5% [33]. In addition, 76.2% of women in India were willing to be screened if provided free of charge, compared to 39.3% of women in our study who were willing to have the Pap smear test, with the most common barrier being the examination cost. The aforementioned barrier could be explained by a lack of a national program to cover the test costs, as Jerash camp is the poorest of Jordan’s ten Palestine refugee camps [20]. Moreover, there was an association between the willingness to undergo a Pap smear test and a woman’s age and marital status. These findings are consistent with what was observed in Nepal, where older and married women had a more positive attitude toward CC screening [34]. Since screening for CC has the potential to dramatically reduce cancer rates in women over the age of 65, who account for almost a fifth of all cervical cancer cases and have higher mortality rates than younger women, it is advised to undergo CC screening around and through menopause [35]. Despite this fact, women after menopause in our study were less likely to have a Pap smear than women before menopause.

In our study, the most recognized symptom was a constant pain in the lower back, followed by bleeding after menopause and having more menstrual periods than usual. These findings are consistent with a survey conducted in Libya, where vaginal bleeding between periods was the most frequently reported symptom [23,36]. Our study revealed that women’s primary sources of information regarding cervical cancer were awareness campaigns, including radio, advertisements, and social networking sites, which is similar to a study on women in the southern region of Saudi Arabia, where the main source of information was social media [37]. This finding concerns us since the content on social media comes from people whose authenticity or reliability is frequently unknown and challenging to evaluate.

Our results suggest that work needs to be performed to increase CC screening awareness and uptake through a strong collaboration between the Jordanian government and UNRWA. The necessary steps needed to improve awareness include initiating awareness campaigns about CC screening that target all age and social groups, implementing a free screening test center, as the cost of the test was the highest reported barrier to being tested, and encouraging health care providers to educate women on being tested routinely. Moreover, a multimedia campaign such as the one the King Hussein Cancer Center (KHCC) holds every year for breast cancer screening awareness would be beneficial. Future research is needed after awareness campaigns to evaluate the effectiveness of the intervention and awareness plans. In a recent meta-analysis, it was shown that self-collected urinary HPV tests have similar accuracy to the cervical HPV test in detecting CIN2 or worse disease stages [38]. Interestingly, the sensitivity of urine HPV was higher in low- and mid-income countries [38]. This highlights the necessity of adopting it as a screening technique for the early diagnosis of cervical pre-cancer, particularly in marginalized populations with limited access to screening units.

The strength of our study is that it addresses a sensitive topic among the refugees, who are considered a conservative community that cannot be easily reached in a different way. In addition, the large sample size and high participation rate. Moreover, using a validated questionnaire and the self-administered questionnaires reduced interviewer bias. On the contrary, it has some limitations; a non-random convenience sampling method was used to find participants due to the strict security regulations in the refugee camps. Lastly, our study did not assess the participants’ attitudes towards taking the HPV vaccine, which is an important factor.

## 5. Conclusions

Our study reveals that the knowledge about CC risk factors in Jerash camp’s refugees was fair, but the knowledge about HPV and the willingness to undergo Pap test screening in the future was very low. This emphasizes the relevance of our research in raising CC awareness, especially among the refugee community as a marginalized community, and it directs health officials to adopt new methods and recommendations to encourage Pap smear testing and HPV vaccination.

## Figures and Tables

**Table 1 healthcare-10-01343-t001:** Socio-demographics and Pap smear screening characteristic of participants.

Demographics Characteristics	Mean (SD) or n (%)
Mean age (SD)	38.99 ± 10.53 (range, 19–64)
Mean marriage age (SD)	21.34 ± 5.16 (range, 13–50)
Marital status	
Married	280 (78)
Single	40 (11.1)
Divorced	21 (5.8)
Widow	18 (5)
Number of children	
0	66 (18.4)
1–3	90 (25.1)
4–6	135 (37.6)
≥7	68 (18.9)
Level of education	
Illiterate	5 (1.4)
Primary school	104 (29)
Secondary school	145 (40.4)
Tertiary education	105 (29.2)
Employment status	
Unemployed	305 (85)
Employee	54 (15)
Family monthly income ^a^	
No income	34 (9.5)
1–200	156 (43.5)
201–400	94 (26.2)
>400	45 (12.5)
Unstable income	18 (4.9)
Prefer not to say	12 (3.3)
Insurance	
Yes	79 (22)
No	280 (78)
**Pap smear screening characteristics**	
Age to start Pap smear	32.5 ± 7.89 (range, 21–50)
Number of Pap smear tests since marriage	1.48 ± 0.79 (range, 0–4)
Pap smear screening frequency (Years)	6.14 ± 4.83 (range, 1–15)

Tertiary education includes community college diploma, bachelor’s degree, master’s, or doctorate; ^a^ Monthly household income in JOD = Jordanian Dinar; 1 JOD = 1.41 USD.

**Table 2 healthcare-10-01343-t002:** Knowledge and belief of participants about symptoms and signs of cervical cancer.

Symptoms and Signs	Yes	No	I Don’t Know
Bleeding between periods	159 (44.3)	150 (41.8)	50 (13.9)
Constant pain in the lower back	177 (49.3)	131 (36.5)	51 (14.2)
Persistent, foul-smelling vaginal discharge	160 (44.6)	154 (42.9)	45 (12.5)
More days of menstruation than usual	166 (46.2)	146 (40.7)	47 (13.1)
Persistent diarrhea	43 (12)	240 (66.9)	76 (21.2)
Bleeding after menopause	171 (47.6)	140 (39)	48 (13.4)
Persistent pelvic pain	164 (45.7)	147 (40.9)	48 (13.4)
Presence of blood in urine or stool	105 (29.2)	185 (51.5)	69 (19.2)
Sudden and unexplained weight loss	158 (44)	142 (39.6)	59 (16.4)
Increased urination	93 (25.9)	187 (52.1)	79 (22)

**Table 3 healthcare-10-01343-t003:** Knowledge of participants on human papillomavirus.

Statements	True	False	I Don’t Know
HPV is rare and has a low incidence	24 (49)	14 (28.6)	11 (22.4)
HPV signs and symptoms always appear when infected	20 (40.8)	15 (30.6)	14 (28.6)
HPV can cause cervical cancer	28 (57.1)	9 (18.4)	12 (24.5)
HPV can be treated with antibiotics	19 (38.8)	15 (30.6)	15 (30.6)
Early marriage increases the possibility of infection with HPV	13 (26.5)	18 (36.7)	18 (36.7)

Only 13.6% of participants who answered that they heard about HPV were able to answer these questions; HPV, Human papillomavirus.

**Table 4 healthcare-10-01343-t004:** Knowledge of risk factors for cervical cancer among participants.

Statements	S D	D	N	A	S A
HPV infection increase the incidence of CC	7 (1.9)	20 (5.6)	234 (65.2)	74 (20.6)	24 (6.7)
Smoking of increase the incidence of CC	17 (4.7)	54 (15)	73 (20.3)	142 (39.6)	73 (20.3)
Weak immune system increase the risk of CC	5 (1.4)	34 (9.5)	90 (25.1)	157 (43.7)	73 (20.3)
Long-term use of contraceptive pills increase the risk of CC	14 (3.9)	45 (12.5)	100 (27.9)	147 (40.9)	53 (14.8)
Chlamydia bacteria infection increase the risk of CC	8 (2.2)	27 (7.5)	186 (51.8)	102 (28.4)	36 (10)
Early marriage (under 17 years old) increase the risk of CC	34 (9.5)	91 (25.3)	128 (35.7)	78 (21.7)	28 (7.8)
The more children I have, the higher the risk of CC	56 (15.6)	117 (32.6)	106 (29.5)	62 (17.3)	18 (5)
Your husband’s relationship with another wife, now or in the past, increases your risk of CC	33 (9.2)	91 (25.3)	116 (32.3)	86 (24)	33 (9.2)
Not having regular Pap smears increase the risk of CC	8 (2.2)	51 (14.2)	113 (31.5)	155 (43.2)	32 (8.9)
Female between the age of 30 and 65 more likely to develop CC	20 (5.6)	38 (10.6)	115 (32)	149 (41.5)	37 (10.3)
Increasing the number of abortions increase the risk of CC	23 (6.4)	48 (13.4)	136 (37.9)	122 (34)	30 (8.4)

Data are represented in n (%); S A, strongly agree; A, agree; N, neutral; D, disagree; S D, strongly disagree; HPV, Human papillomavirus; CC, Cervical Cancer.

**Table 5 healthcare-10-01343-t005:** Knowledge of participants of the Pap smear test (n = 264).

Statements	True	False	I Don’t Know
The testing area must be clean before the examination begins	211 (79.9)	43 (16.3)	10 (3.8)
All women of childbearing age should undergo a test once every 5 years	157 (59.5)	47 (17.8)	60 (22.7)
The test is only for women at risk of cervical cancer	58 (22)	172 (65.2)	34 (12.9)

**Table 6 healthcare-10-01343-t006:** Participants’ attitudes toward refusing routine cervical cancer screening.

Statements	Yes, It Prevents Me Somewhat	Yes, It Strongly Prevents Me	No, It Has No Effect
Feeling ashamed and shy about doing this test	127 (35.4)	55 (15.3)	177 (49.3)
My cultural and religious beliefs prohibit this test	78 (21.7)	19 (5.3)	262 (73)
My health is good; I do not need to do the test	140 (39)	69 (19.2)	150 (41.8)
Lack of health services in the area where I live	129 (35.9)	103 (28.7)	127 (35.4)
Bad behavior of health personnel with me previously	101 (28.1)	56 (15.6)	202 (56.3)
I think the test will hurt me, so I avoid it	121 (33.7)	60 (16.7)	178 (49.6)
I have already undergone the test, and it hurts	74 (20.6)	33 (9.2)	252 (70.2)
Costs of the examination	112 (31.2)	130 (36.2)	117 (32.6)
If the test was performed for me by a male doctor	99 (27.6)	140 (39)	120 (33.4)
Fear of knowing the result	110 (30.6)	92 (25.6)	157 (43.7)

## Data Availability

The data from the present research that were utilized and analyzed are accessible from the corresponding author upon reasonable request.

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
