# Peer review of "Cervical Cancer Screening among Female Refugees in Jordan: A Cross-Sectional Study"

_healthcare, 2022, doi:10.3390/healthcare10071343_

Round 1

Reviewer 1 Report

This document is important because it examines for the first time the awareness of screening methods to detect the infection with high-risk oncogenic HPV and to have routine gynaecological examination,  in refugee  Palestinians women of camps in Jordan compared  with other studies already conduted in native  Jordanian  women.
This study reveals that the knowledge about cervical cancer risk factors in this women was low and the knowledge about HPV  and the willingness to do Pap-test screening in the future was very low. For this  marginalised community, this study is relevant, because directs health officials to adopt methods and recommendations to implement Pap smear testing probably HPV test in urine or self-colected samples (the Authors should refer this two methods )  and HPV vaccination.

Author Response

Dear Reviewer,

We thank you for your meticulous comments and tireless efforts in trying to direct us to make our manuscript more concrete and of higher quality. All of your comments and/or suggested changes were addressed thoroughly in the responses below. We appreciate your insightful perspectives.

Comment 1: “This document is important because it examines for the first time the awareness of screening methods to detect the infection with high-risk oncogenic HPV and to have routine gynaecological examination,  in refugee Palestinian women of camps in Jordan compared  with other studies already conduted on native Jordanian women.

This study reveals that the knowledge about cervical cancer risk factors in this population was low and the knowledge about HPV and the willingness to do Pap-test screening in the future was very low. For this  marginalised community, this study is relevant, because it directs health officials to adopt methods and recommendations to implement Pap smear testing probably HPV test in urine or self-collected samples (the Authors should refer this two methods )  and HPV vaccination.”

Response: We thank the editor for his/her comment. The following paragraph was added into the discussion, as requested. “In a recent meta-analysis, it was shown that self-collected urinary HPV tests have similar accuracy to the cervical HPV test to detect CIN2 or worse disease stages [38]. Interestingly, the sensitivity of urine HPV was higher in low- and mid-income countries [38]. This highlights the necessity of adopting it as a screening technique for the early diagnosis of cervical pre-cancer, particularly in marginalized populations with limited access to screening units.”

Reviewer 2 Report

Thank you for the opportunity to review the work by Nadia Muhaidat et. Al. The study well-structured and designed and is really interesting. The paper presents a socially important topic. I believe that it is valuable both for the readers and also the government should pay attention to it. The work requires a few minor corrections.

1.     Minor typos 2.     The introduction mentions the recommendations of the American Cancer Society, but nowhere does it say that HPV screening is recommended for testing or co-testing rather than a Pap smear. 3.     The exclusion criteria include patients who underwent a hysterectomy; it is not described later how many were rejected for this reason. Are there reports or other papers that mention such a criterion? 4.     Sample size and sampling procedures” - the text is repeated in the next section "data collection procedures". Please correct it.

5.      Please standardize the terminology; the authors use “Pap smear” once, “pap smear” another time.

Author Response

Dear Reviewer,

We thank you for your meticulous comments and tireless efforts in trying to direct us to make our manuscript more concrete and of higher quality. All of your comments and/or suggested changes were addressed thoroughly in the responses below. We appreciate your insightful perspectives.

Comment 1: “Thank you for the opportunity to review the work by Nadia Muhaidat et. Al. The study well-structured and designed and is really interesting. The paper presents a socially important topic. I believe that it is valuable both for the readers and also the government should pay attention to it. The work requires a few minor corrections.”

Response: We thank the editor for his/her comment. All of your comments and/or suggested changes were addressed thoroughly in the responses below.

Comment 2: “Minor typos”

Response: Accordingly, the manuscript was sent to two English experts for language editing.

Comment 3: “The introduction mentions the recommendations of the American Cancer Society, but nowhere does it say that HPV screening is recommended for testing or co-testing rather than a Pap smear.”

Response: We agree with the reviewer on the importance of the mentioned point. The following paragraph was added into the introduction. “Co-testing with Pap smear and HPV testing every five years is recommended for women aged 30 to 65 due to its better sensitivity, whereas Pap smear testing is indicated for women aged 21 to 30 due to HPV testing's lower specificity in this population [5].”

Comment 4: “The exclusion criteria include patients who underwent a hysterectomy; it is not described later how many were rejected for this reason. Are there reports or other papers that mention such a criterion?”

Response: The criteria used were adopted after an extensive literature review and discussing it with a group of consultants from the Department of Gynecology. Here are some published articles that mentioned hysterectomy as an exclusion criteria (DOI: 10.1155/2020/9690473, DOI: 10.1177/1090198117742153).

Comment 5: ““Sample size and sampling procedures” - the text is repeated in the next section "data collection procedures". Please correct it.”

Response: We thank the reviewer for his/her comment and apologize for this mistake. It has been changed accordingly.

Comment 6: “Please standardize the terminology; the authors use “Pap smear” once, “pap smear” another time”

Response: We apologize for this mistake. It has been changed accordingly. In addition, the manuscript was sent to two English experts for language editing

Reviewer 3 Report

Comments to the Authors:

This is an interesting and overall well-written paper, this manuscript finds that refugee women have a fair level of knowledge of cervical cancer risk factors but they lack the motivation to have a Pap test to screen for the disease by a cross-sectional study. And effort should be made to raise awareness about the issue and promote testing for underrepresented women in refugee camps. It will be a solid contribution to the Healthcare and will certainly appeal to many of its readers. I address some of the main issues with the manuscript in the next few paragraphs. It is recommended that this manuscript be published in Healthcare after completing minor revision.

1.      Please keep the tense consistent throughout the text and correct writing mistakes, such as line 29 “cab” should be “can”.

2.      There are two repeated paragraphs in the 2.2 and 2.3, please check and delete one of them.

3.      In the third paragraph of 2.3, “Data collection procedures” has different format from other titles, please correct it.

4.      There are not lots of evidences to support the discussion, please enrich data to support the discussion. For example, I can not learn that there is a relation between the knowledge of cervical cancer and education.

5.      The introduction mainly introduces the knowledge of cervical cancer. It should add some anticancer therapies to support the article. So the following recently published important related papers should be cited: Chem. Soc. Rev. 2017, 46, 7021; Chem. Soc. Rev. 2021, 50, 2839; Adv Mater. 2022, 34, 2106388.

Author Response

Dear Reviewer,

We thank you for your meticulous comments and tireless efforts in trying to direct us to make our manuscript more concrete and of higher quality. All of your comments and/or suggested changes were addressed thoroughly in the responses below. We appreciate your insightful perspectives.

Comment 1: “This is an interesting and overall well-written paper, this manuscript finds that refugee women have a fair level of knowledge of cervical cancer risk factors but they lack the motivation to have a Pap test to screen for the disease by a cross-sectional study. And effort should be made to raise awareness about the issue and promote testing for underrepresented women in refugee camps. It will be a solid contribution to the Healthcare and will certainly appeal to many of its readers. I address some of the main issues with the manuscript in the next few paragraphs. It is recommended that this manuscript be published in Healthcare after completing minor revision.”

Response: We thank the editor for his/her comment. All of your comments and/or suggested changes were addressed thoroughly in the responses below.

Comment 2: “Please keep the tense consistent throughout the text and correct writing mistakes, such as line 29 “cab” should be “can”.”

Response: We apologize for this mistake. It has been changed accordingly. In addition, the manuscript was sent to two English experts for language editing.

Comment 3: “There are two repeated paragraphs in the 2.2 and 2.3, please check and delete one of them.”

Response: We thank the reviewer for his/her comment and apologize for this mistake. It has been changed accordingly.

Comment 4: “In the third paragraph of 2.3, “Data collection procedures” has different format from other titles, please correct it.”

Response: We have changed the format to match the journal’s guidelines.

Comment 5: “There are not lots of evidences to support the discussion, please enrich data to support the discussion. For example, I can not learn that there is a relation between the knowledge of cervical cancer and education.”

Response: We have added the following paragraphs to the discussion to make it clearer “Numerous studies have shown that education level has a significant impact on screening uptake and women's knowledge of CC risk factors. A study about the uptake of cervical and breast cancer screening among women in Spain found that those with higher levels of education were more likely to have undergone cervical cancer screening than those with lower levels of education or only a primary education [27]. In addition, prior cross-sectional research among women in the Gaza Strip revealed that participants with only a secondary or diploma degree had a reduced likelihood of being well-versed in CC risk factors [25,28].”, “Where it was found that the overall awareness of CC risk factors with good knowledge was 23.7%, which is similar to our study, which found it to be 18.4%. [25]” and “Moreover, more than half (57.1%) of those who had heard of HPV knew that it is a direct cause of cervical cancer, which is more than the knowledge level reported in a study among Chinese women, which was found to be 53.4% [31].”

Comment 6: “The introduction mainly introduces the knowledge of cervical cancer. It should add some anticancer therapies to support the article. So the following recently published important related papers should be cited: Chem. Soc. Rev. 2017, 46, 7021; Chem. Soc. Rev. 2021, 50, 2839; Adv Mater. 2022, 34, 2106388”

Response: We agree with the reviewer on the importance of the mentioned point. The following paragraph was added into the introduction. “The National Comprehensive Cancer Network (NCCN) 2019 guidelines describe early-stage disease (stages I–II) treatment options including surgery and concurrent chemoradiation, which can result in cures in 80% of patients [13]. According to the International Federation of Gynecology and Obstetrics (FIGO), treatments are dependent on the stage, and surgery is usually only used for early-stage disease, fertility-preservation, and smaller lesions, such as stage IA, IB1, and selected IIA1 cases [14]. Cervical cancer stages IB–IIA can be treated with either radical surgery or radiotherapy, and the survival outcomes are the same for both [15]. In more advanced cases, concurrent chemoradiation using cisplatin alone or a cisplatin and fluorouracil combination is the treatment of choice for stages IB2, II, III, and IVA disease [13].”